# Generation of Functional Cardiomyocytes from Human Gastric Fibroblast-Derived Induced Pluripotent Stem Cells

**DOI:** 10.3390/biomedicines9111565

**Published:** 2021-10-29

**Authors:** Chih-Hsien Wu, Hsuan-Hwai Lin, Yi-Ying Wu, Yi-Lin Chiu, Li-Yen Huang, Cheng-Chung Cheng, Chung-Chi Yang, Tsung-Neng Tsai

**Affiliations:** 1Department of Biochemistry, National Defense Medical Center, Taipei 11490, Taiwan; claudia_csmu13@hotmail.com (C.-H.W.); yc566@georgetown.edu (Y.-L.C.); 2Division of Cardiology, Department of Internal Medicine, Tri-Service General Hospital, National Defense Medical Center, Taipei 11490, Taiwan; vinecristine@yahoo.com.tw (L.-Y.H.); chengcc@mail.ndmctsgh.edu.tw (C.-C.C.); t220979@gmail.com (C.-C.Y.); 3Division of Gastroenterology, Department of Internal Medicine, Tri-Service General Hospital, National Defense Medical Centre, Taipei 11490, Taiwan; redstone120@gmail.com; 4Division of Hematology/Oncology, Department of Medicine, Tri-Service General Hospital, National Defense Medical Center, Taipei 11490, Taiwan; rq0922@gmail.com; 5Division of Cardiology, Department of Internal Medicine, Taoyuan Armed Forces General Hospital, Taoyuan County 32551, Taiwan

**Keywords:** induced pluripotent stem cells, human gastric fibroblast cells, cardiomyocytes

## Abstract

Coronary artery diseases are major problems of the world. Coronary artery disease patients frequently suffer from peptic ulcers when they receive aspirin treatment. For diagnostic and therapeutic purposes, the implementation of panendoscopy (PES) with biopsy is necessary. Some biopsy samples are wasted after the assay is completed. In the present study, we established a protocol for human gastric fibroblast isolation and induced pluripotent stem cell (iPSC) generation from gastric fibroblasts via PES with biopsy. We showed that these iPSCs can be differentiated into functional cardiomyocytes in vitro. To our knowledge, this is the first study to generate iPSCs from gastric fibroblasts in vitro.

## 1. Introduction

Coronary artery disease (CAD) is a major cause of death worldwide. Aspirin has been used as a first-line treatment for CAD patients according to the American Heart Association and European Society of Cardiology guidelines. However, aspirin could lead to peptic ulcers and frequent peptic ulcer bleeding. The coincidence of coronary artery disease and peptic ulcers occurs frequently, especially in patients receiving aspirin medication. The progression of CAD can lead to ischemic cardiomyopathy and eventually heart failure. Half of these patients will die within 5 years due to poor life quality. Moreover, these patients have a high risk for upper gastrointestinal (GI) bleeding, which requires panendoscopy (PES) for diagnosis and treatment. In the process of PES, a gastric biopsy will be performed to exclude the possibility of gastric cancer and Helicobacter pylori infection. Some gastric samples are discarded after the assay is completed. Human gastric fibroblast cells (hGFCs), a type of mesenchymal cell rising from the embryonic mesoderm, participate in gastric wound healing. Gastric fibroblasts can secrete proteins, which are the main sources of extracellular matrix of the stomach. They also play a critical role in regulating epithelial cell activities, such as proliferation and restitution [1]. Somatic cells have been reprogrammed by introducing four genes to revert them into induced pluripotent stem cells (iPSCs), which provide an alternative to using embryonic stem cells (ESCs) [2,3]. These cells share numerous characteristics with ESCs, such as self-renewal to persist in their undifferentiated state and differentiation potential to specialized cell types, but without the technical challenge and complicated ethical and legal concerns. These findings suggest that iPSCs are not only an advancement in disease research but also an important foundation for the development of autologous cell therapies. Several groups showed that iPSCs can be generated from different cell types of mice, humans, and other species. Moreover, studies demonstrated that iPSCs can be differentiated into cardiomyocytes. The low efficacy of reprogrammed cells is one of the obstacles of iPSC generation. Due to pluripotency, proliferative capacities, and transcriptomes, fibroblasts could be a potential cell source for iPSC generation. However, fibroblasts might be differently obtained from tissue. Previous reports showed that skin fibroblasts can be reprogrammed into iPSCs. hGFCs play a key role in the development and progression of gastric carcinoma [4]; however, no report has demonstrated the successful reprogramming of human gastric fibroblasts (hGFs).

Most cell lines or primary culture cells are obtained from cancer patients who have received gastrectomy. These cells can still possess tumor characteristics [5]. However, the cells obtained from biopsy tissues lack the numbers for multiple cell culture experiments. Three major types of fibers, including collagen, elastic, and reticular fibers, are secreted from fibroblasts, and they are rich in type I and III collagen [6]. Collagenases are applied as suitable enzymes for the separation of tissue cells in medical investigations, especially for tiny-tissue cell collection. In the present study, we established a new protocol for isolating hGFCs from tiny gastric tissue and their iPSC generation.

## 2. Materials and Methods

### 2.1. Ethics Statement

After the approval of the study protocol by the ethics committee of the Institutional Review Board, Tri-Service General Hospital (NO.1-104-05-116, Taiwan) and obtaining written informed consent, human gastric tissues were obtained from the stomach by PES with biopsy.

### 2.2. Cell Isolation and Culture

Gastric tissues and skin were incubated with 0.2 and 0.8 PZ U/mltype IV collagenase (SERVA Electrophoresis, Nordmark GmBH, Crescent Chemical, Heidelberg, Germany) for 1 h and 2 h at 37 °C, respectively, followed by washing with phosphate-buffered saline (PBS). Then, the hGFC cells were seeded in a six-well plate with 10% FBS (Gibco, Waltham, MA, USA), RPMI 1640 (Gibco, Grand Island, NY, USA), 10% FBS DMEM, or FibroLife^®^ Serum-Free Medium Complete Kit (LifeLine Cell Technology, Frederick, MD, USA). The media were refreshed every other day and passaged when 70% confluence was reached. The cells from passages 6 to 12 were used for the experiment. In addition, the skin fibroblasts were cultured with 10% FBS RPMI1 1640. BJ cells were obtained from American Type Culture Collection (Manassas, VA, USA) and cultured with 10% FBS MEM (Gibco, Grand Island, NY, USA). The cells were passaged when 70% confluence was reached.

For iPSC culture, the cells were passaged every 5 days at a split ratio of 1:6 using 5 mM ethylenediaminetetraacetic acid (EDTA) (Thermo Fisher Scientific, Waltham, MA, USA) on Geltrex™ LDEV-Free Reduced Growth Factor Basement Membrane Matrix (Gibco, Waltham, MA, USA)-coated culture dishes. StemFlex^TM^ medium (Gibco, Waltham, MA, USA) was used to maintain and expand the colonies of iPSCs. All cells were cultured at 37 °C in an atmosphere of 5% CO_2_.

### 2.3. hGFC Reprogramming

hGFCs were seeded on a six-well plate and cultured with 10% FBS RPMI1640. After 70% confluency was reached, the cells were collected and transfected with OCT3/4, SOX2, KLF4, and c-Myc using CytoTune^®^-iPS 2.0 Sendai Reprogramming Kit (Thermo Fisher Scientific, Waltham, MA, USA) at the multiplicity of infection of 5:5:3 [3]. Then, 2 × 10^5^ transfected cells were seeded on a six-well plate coated with Geltrex™ LDEV-Free Reduced Growth Factor Basement Membrane Matrix on the next day. The fibroblast medium (10% FBS RPMI1640) was replaced one day after transfection and every other day thereafter. About 8 days after transduction, the culture medium was replaced with Essential 8™ Medium (Gibco, Grand Island, NY, USA) and replaced daily. Within 15–21 days of transfection, the colonies were picked for expansion as individual iPSC lines. After passage 10, the iPSCs were cultured with StemFlex medium to determine the rate of expansion. Passages 20–30 were used for the experiment.

### 2.4. WST-1 Assay

To investigate the cell variability and proliferation after incubating the hGFCs with the designed medium, the WST-1 assay was performed. The hGFCs (1 × 10^4^ cells/well) were seeded in a 96-well plate and cultured with 100 μL 10% FBS RPMI1640, 10% FBS DMEM, or FibroLife serum-free medium at 37 °C in 5% CO_2_ incubator for 72 h. A total of 10 µL WST-1 reagent (Abcam, Cambridge, MA, USA) was added to each well, and the cells were incubated at 37 °C for 2 h. The optical density values were determined using a microplate reader at 450 nm. Each condition was performed in triplicate.

### 2.5. Wound-Healing Assay

Wound healing was assayed using 2-well silicone inserts (ibidi GmbH, Gräfelfing, Germany) placed into a 12-well plate. A total of 5 × 104 cells/well were plated with 70 µL 10% FBS medium. The cell culture inserts were removed after 24 h, leaving a defined cell-free gap of 500 μm. A total of 0.5 mL of 10% FBS medium was added into each well of the 12-well plate at 0 h and images were taken at 40× magnification. The cell numbers at 16 h were determined by using image analysis software (FIJI, ImageJ V1.52p (National Institutes of Health, Bethesda, MD, USA). Data represent means ± SD from three independent experiments performed.

### 2.6. RNA Extraction and RT-PCR

Total RNA was extracted with TRIzol™ Reagent (Thermo Fisher Scientific, 15596026) as per the manufacturer’s instructions. RT was carried out with 2 µg RNA, random primers, and Moloney murine leukemia virus reverse transcriptase (Promega, M170, Madison, WI, USA) at a final volume of 20 µL. SYBR Green quantitative PCR (qPCR) was performed on a Bio-Rad CFX96^TM^ real-time system with 96-well optical reaction plates. Each 10 µL SYBR Green qPCR contained iTaq Universal SYBR^®^ Green Supermix (Bio-Rad, 1725121, Hercules, CA, USA), 20 ng cDNA, and 5 µM of each specific primer. For calponin, 5′-ATGTCCTCTGCTCACTTCAAC-3′ (forward) and 5′-CACGTTCACCTTGTTTCCTTTC-3′ (reverse); for CD90, 5′-GAAGGTCCTCTACTTATCCGCC-3′ (forward) and 5′-TGATGCCCTCACACTTGACCAG-3′ (reverse); for VE-cadherin, 5′-GAAGCCTCTGATTGGCACAGTG-3′ (forward) and 5′-TTTTGTGACTCGGAA GAACTGGC-3′ (reverse); for GAPDH, 5′-CATCACTGCCACCCAGAAGACTG-3′ (forward) and 5′-ATGCCAGTGAGCTTCCCGTTCAG-3′ (reverse). The specificity of each assay was validated by dissociation curve analysis. The combined results were from at least three independent experiments, wherein each sample was assayed in duplicate, and a mean value was used for the relative quantification of mRNA levels by the comparative Cq method with β-actin as the reference gene.

### 2.7. Immunofluorescence Staining

Immunostaining was performed using anti-OCT4, anti-SEEA4, anti-TRA-1-60, anti-SOX2 (Thermo Fisher Scientific, Waltham, MA, USA), anti-NKX2.5, anti-TNNT2/cTNT, anti-alpha-actinin (Sigma-Aldrich, St. Louis, MO, USA, A7811), and MYL2 (ProteinTech, Rosemont, IL, USA, 10906-1-AP) as primary antibodies. The procedure was performed on a 24-well culture dish with 60–70% cell confluence. The cells were washed with PBS twice, followed by fixing with 4% formaldehyde in Dulbecco’s PBS (DPBS) for 15 min. After additional washing with PBS, the cells were permeabilized by 1% saponin in DPBS for 5 min and blocked with 3% BSA in DPBS for 30 min at room temperature. The cells were incubated with the designed primary antibody overnight at 4 °C. Then, the cells were washed thrice with PBS and incubated with the designed secondary antibodies for 1 h. The chamber was washed thrice with PBS, and nuclear counterstaining was carried out with NucBlue™ Fixed Cell ReadyProbes™ Reagent (4′,6-diamidino-2-phenylindole (DAPI), Thermo Fisher Scientific, Waltham, MA, USA) for 1 min. After washing with PBS twice, images were captured with a Zeiss Axioplan 2 Imaging microscope with Plan-NEOPLUAR 10×/0.75 objective lens.

### 2.8. Western Blot Assay

The Western blot assay had been performed in our previous work [7]. Protein was collected by scraping and carried out on ice in RIPA buffer and a protease-inhibiting cocktail (Complete Roche Molecular Biochemi-cals, Almere, The Netherlands). Then, the samples were run on 10% SDS-PAGE, electroblotted onto membranes, and incubated with primary antibodies against human CD90 (13801, Cell Signaling Technology, Danvers, MA, USA), VE-cadherin (MA5-17050, Thermo Fisher Scientific, Waltham, MA, USA), and GAPDH (PA1-987; Thermo Fisher Scientific, Waltham, MA, USA). The expression of designed proteins was observed with enhanced chemiluminescence using peroxidase-labeled luminol as the detection fluid (ECL; Amersham Life Sciences, Little Chalfont, UK).

### 2.9. ALP Live Staining

The cell culture medium was removed and washed with DMEM/F-12 for 3 min twice. The cells were cultured with DMEM/F12 medium containing ALP live staining solution (Thermo Fisher Scientific, Waltham, MA, USA) for 30 min, followed by washing with DMEM/F12 medium for 5 min twice. The image was captured immediately with a Zeiss Axioplan 2 Imaging microscope with Plan-NEOPLUAR 10×/0.75 objective lens.

### 2.10. Teratoma Formation Assay

Nonobese diabetic/severe combined immunodeficiency (NOD/SCID) mice were obtained from BioLASCO Taiwan Co., Ltd., and the experiment was conducted under the approval of (IACUC-15-342). A total of 5 × 10^6^ hGFC-derived iPSCs were injected intramuscularly into the NOD/SCID mice. The hGFC-derived iPSC line at a passage greater than 20 was used. Three mice were injected with the hGFC-derived iPSCs, and three tumors were generated.

### 2.11. H&E Staining

The tissues were collected and fixed in 10% neutral-buffered formalin, cut into small pieces, decalcified in 0.5 M EDTA (pH 8.0) for 7 days, paraffin-embedded, and sectioned. After dehydration, the tissue sections were immersed in hematoxylin for 5 min, washed with tap water for 15 min, quickly dipped in 1% acid alcohol, washed with tap water for 15 min, immersed in 70% ethanol for 3 min, immersed in eosin for 1 min, washed with tap water for 15 min, and dehydrated by immersion in a series of increasing concentrations of alcohol (twice in 95% concentration for 2 min and twice in 100% concentration for 2 min). The sections were then immersed in xylene twice for 5 min each and mounted with mounting media (Permount, Fisher Chemical, Fair Lawn, NJ, USA).

### 2.12. Karyotyping Analysis

To evaluate the genetic stability of the hGFC-derived iPSCs, karyotype analyses were performed by Ko’s Obstetrics and Gynecology Clinic Laboratories (Taipei, Taiwan) using cytogenetic techniques and analyzed by high-resolution G-banding.

### 2.13. Cardiomyocyte Differentiation

iPSCs were seeded in a 12-well plate and cultured until 60% confluence was reached. The Cardiomyocyte Differentiation Medium A (Thermo Fisher Scientific, Waltham, MA, USA) was added and replaced with Medium B two days later. After incubation with Medium B (Thermo Fisher Scientific, Waltham, MA, USA) for another 2 days, Cardiomyocyte Maintenance Medium (Thermo Fisher Scientific, Waltham, MA, USA) was added and replaced daily until the cells were collected.

### 2.14. Intracellular Ca^2+^ Concentration Measurement

hGFC-derived iPSCs were seeded on coverslips and cultured with cardiomyocyte differentiation medium for 14 days. The change in Ca^2+^ concentration in single cells was then measured using fluorescent dye fura-2. After incubation with 5 μM fura-2 acetoxymethyl ester in Locke’s buffer consisting of (in mM) 150 NaCl, 5 KCl, 1 MgCl_2_, 2.2 CaCl_2_, 5 glucose, and 10 HEPES, pH7.4 at 37 °C for 60 min, cells were then washed twice and further incubated at 37 °C for 30 min in Locke’s buffer. Cells were then bathed in Locke’s buffer in order to measure 50 mM KCl-induced Ca^2+^ increase. The cells were treated with phosphate-buffered saline as control. The coverslips were mounted in a modified Cunningham chamber attached to the stage of a fluorescence microscope (model DMIRB; Leica, Mannheim, Germany) that was equipped with a high-speed scanning polychromatic light source (model C7773; Hamamatsu Photonics, Shizuoka, Japan) and a CCD camera (HISCA; model C6790; Hamamatsu Photonics) controlled by Aquacosmos 2.5 software (Hamamatsu Photonics). Quantitative fluorescence intensity data were acquired at excitation wavelengths of 340 and 380 nm and an emission wavelength of 505 nm; a sampling rate of 1 Hz was used. The 340 to 380 nm fluorescence ratio (F340/F380) was used to reflect intracellular Ca^2+^ changes. All experiments were performed five times and similar results were obtained. Results from one representative experiment are illustrated graphically.

### 2.15. Statistical Analysis

The results were analyzed using the SPSS version 17 statistics software (SPSS Inc., Chicago, IL, USA), which was used for data analysis. To investigate the difference in results, ANOVA and two-tailed Student’s t-test were used for comparisons. A *p*-value < 0.05 was indicated as statistically significant.

## 3. Results

### 3.1. Isolation and Characterization of hGFCs

Figure 1A presents the hGFC isolation and culture. The stomach tissue was obtained by biopsy when PES was performed. After digestion with collagenase IV, the cells were cultured with 10% fetal bovine serum–Roswell Park Memorial Institute 1640 (FBS RPMI1640), 10% FBS–Dulbecco’s Modified Eagle Medium (DMEM), or FibroLife serum-free medium. The cells presented a typical spindle shape, and cells from passages 6 to 12 were used for subsequent studies due to the fact that other kinds of cells would not survive in these culture conditions and time points [8]. The cells were cultured with 10% FBS RPMI1640, 10% FBS DMEM, or FibroLife serum-free medium to identify the medium suitable for cell culture. WST-1 assay was performed 72 h post-incubation to measure the cell viability. The data showed that cells cultured with 10% FBS RPMI1640 or 10% FBS DMEM were more viable than those cultured with FibroLife serum-free medium (Figure 1B). To analyze whether these cells were fibroblasts, we performed reverse-transcription–polymerase chain reaction (RT-PCR) and Western blot to detect the expression of fibroblast markers (*α*-*smooth muscle actin* (α-SMA) and calponin) and epithelium cell markers (vascular endothelial (VE)-cadherin). Smooth muscle cells also present these markers, and thus, the cells were analyzed with CD90 to confirm that the majority of them were fibroblasts. The data showed that these cells expressed α-SMA, calponin, and CD90 as part of the fibroblast cell line, that is, BJ and skin fibroblast cells. The epithelial gastric cell marker VE-cadherin was absent in these primary and BJ cells (Figure 2A,C, respectively). In addition, the immunofluorescent stain with hGFCs expressed the fibroblast markers α-SMA and calponin (Figure 2B). These results revealed that the primary cells obtained from the gastric tissue may be fibroblasts (hGFCs). In addition, fibroblasts will anticipate wound healing when tissue has been injured. To identify the hGFC migration function compared with skin fibroblasts and BJ cells, the wound healing assay was performed. Our data revealed that hGFCs had less migration activity than the skin fibroblasts and BJ cells (Figure 2D,E). This indicated the variation property between these fibroblast cells.

### 3.2. hGFCs as a Cell Source for iPSC Generation

#### 3.2.1. hGFC-Derived iPSCs Exhibit Similar Morphology and Cell Marker Expression of Human ESCs with Normal Karyotyping

hGFs have the characteristic of mesenchymal cells, which exist as the embryonic mesoderm. These cells secrete the extracellular matrix, which is rich in type I and/or type III collagen. This matrix plays a critical role in stomach wound healing. In addition, gastric fibroblasts are involved in the development and progression of gastric cancer [9]. To verify these cells as a cell source for iPSC generation, we reprogrammed the hGFCs using the CytoTune™-iPS 2.0 Reprogramming System. Fifteen days after reprogramming, the iPSC colonies were picked up, isolated, and passaged (Figure 3A). To examine whether the cells have the same features as ESCs, we characterized the pluripotency of the generated iPSCs by immunofluorescence and alkaline phosphatase (ALP) staining. The results showed that iPSCs revealed a typical ESC morphology and expressed the pluripotency markers TRA-1-60 and SOX2 (Figure 3B). Moreover, the cells displayed high intracellular ALP activity like ESCs (Figure 3C). To assess genomic stability, chromosomal abnormalities, and copy number variants of the iPSC lines, G-band karyotyping was performed. Our data demonstrated that the majority of iPSCs tested had a normal 46, XX karyotype (Figure 3D).

#### 3.2.2. hGFC-Derived iPSCs Formed Teratoma In Vivo

Teratoma formation in immunodeficient mice is essential to define ESCs and iPSCs. This assay showed that the cells can generate a disorganized structure representing the three germ layers [10,11]. The hGFC-derived iPSCs were injected into the mice subcutaneously. After the tumor formed on the injection site, the histological examination with hematoxylin and eosin (H&E) staining revealed advanced differentiation of structures representative of all three embryonic germ layers, including gland-like structures (endoderm) (Figure 4A), cartilage (mesoderm) (Figure 4B), and rosettes of the neural epithelium (ectoderm) (Figure 4C). These results indicated that iPSCs had been reprogrammed into a pluripotent state and can differentiate into the three germ layers.

### 3.3. hGFC-Derived iPSCs Differentiated into Cardiomyocytes In Vitro

To examine whether the hGFC-derived iPSCs differentiated into cardiomyocytes, we propagated the iPSC colonies in the undifferentiated state on the top of the Geltrex-coated layer. Then, the cells were cultured with cardiomyocyte differentiation medium. After 14 days, the beating cardiomyocytes were observed in the culture dish (Appendix A). Immunofluorescence staining was performed to check the cardiac markers of differentiated cardiomyocytes. The results showed that these cells expressed the cardiomyocyte markers NK2 Homeobox 5 (NKX2.5), troponin T type 2 (TNNT2/cTNT), alpha-actinin and Myosin regulatory light chain 2 (MYL2) (Figure 5A–E). Moreover, to show the functionality of iPSC-derived cardiomyocytes, we also measured the intracellular Ca^2+^ change in response to 50 mM KCl treatment, which activates L type Ca^2+^ channels. A marked Ca^2+^ increase was induced by high KCl, while it remained unaltered by phosphate-buffered saline (Figure 5G).

## 4. Discussion

hGFCs represent an abundant population of cells in the stomatic mucosa. These mesenchymal cells are spindle-shaped and plastic-adherent cells and reside beneath the epithelium layer. Gastric fibroblasts are responsible for extracellular matrix synthesis, which plays an important role in gastric wound healing. In addition, hGFCs have been suggested to regulate the differentiation, proliferation, and regeneration of epithelial cells. Research has demonstrated that gastric fibroblasts also participate in the development and progression of gastric carcinoma [9]. Given the effect of these cells, a large number of primary cells are critical for the further exploration of their role in the pathophysiology of gastric diseases. Multiple approaches, including enzymatic tissue digestion, tissue chunks, and enzymatic perfusion of hollow organs, are used for the isolation of fibroblasts from tissues [12]. The efficiencies of fibroblast isolation from various organs are different when using various isolation programs and technologies. Three major types of fibers are secreted from fibroblasts: collagen, elastic, and reticular fibers [13]. Botting et al. compared the cell viability and yields of mononuclear phagocytes isolated from the human dermis using trypsin, collagenase, or collagenase IV. After tissue treatment for 60 min, higher cell viability and yields were observed with collagenase IV than trypsin and collagenase [14]. At present, most of gastric cell isolations were collected from gastric cancer patients who underwent gastrectomy. The cells may contain characteristics of gastric cancer, which limits their use in clinical settings for future iPSC therapy and personalized cell bank setup. Ma et al. reported the use of 0.25% trypsin to digest 5 × 5 × 5 mm^3^ gastric tissues obtained from gastric cancer patients. Then, the cells were cultured with DMEM containing 20% FBS [4]. In the present study, we designed an optimized procedure of enzymatic dissociation with collagenase IV. A 2 × 2 × 1 mm^3^ normal gastric tissue was biopsied from PES and cultured to generate a large number of hGFCs (Figure 1A). Three kinds of medium were used to evaluate the proliferation efficiency of cells (Figure 1B). The results revealed that cells cultured with RPMI1640 and DMEM containing 10% FBS were more proliferative than those cultured with a serum-free culture medium (FibroLife), which provided an effective method to obtain a high number of hGFCs from tiny tissues.

ESCs can differentiate into a variety of cells and are self-renewable. Under appropriate culture conditions, ESCs can divide indefinitely and produce a large number of cells for therapies. However, the availability and ethical issues are major concerns for their use. In 2007, iPSCs were generated from somatic cells through the transduction of defined pluripotency-associated transcription factors [3], showing promise for future stem cell research and potential clinical applications. Similar to ESCs, iPSCs can typically proliferate, self-renew indefinitely, and differentiate into various kinds of cells in vitro. In addition, iPSCs not only greatly reduce immune rejection problems but also minimize ethical considerations. iPSCs have been used to derive a wide variety of cells from different species, such as mice, rats, rabbits, pigs, and humans [15,16,17,18,19]. Several types of human somatic cells, including peripheral blood mononuclear cells, cord blood, amniotic cells, fibroblasts, and hepatocytes, have been subjected to in vitro reprogramming [20,21,22,23,24]. These studies used different protocols on similar cell sources. In 2007, Takahashi et al. reported the first case of the generation of human IPSCs from BJ cells [3]. The BJ cell derived from human neonatal foreskin has a high proliferation rate that minimized the shortage of low efficacy of reprogramming. Then, skin fibroblasts were widely used to generate iPSCs [25,26]. Here, we established a simple and universally applicable method to generate iPSCs from hGFCs. This study is the first to report the use of hGFCs as a cell source to generate iPSCs in vitro. The hGFCs had less proliferation and migration activity compared to the skin fibroblasts and BJ cells. The data revealed that the different tissue residues of fibroblasts will have different activities. The feature of cell markers, pluripotency and proliferative rate, and transcriptomes of mesenchymal stem cells and fibroblasts suggest the similarities between these two types of cell. Fibroblasts retain the de-differentiation potential and pluripotent properties, which could enhance reprogramming capacity. This suggests that fibroblasts remain prime candidates to establish induced pluripotent stem cells (iPSCs). However, the tissue residual fibroblast originated from different gremial layers could have a different plasticity [27]. In addition, Sanchez-Freire et al. reported that the different somatic cells could lead to different capacities to differentiate to specific cells. This study suggested that while epigenetic memory will impact the differentiation efficiency of the somatic cell source, the skin is derived from the ectodermal layer. However, the gastric connective tissue originated from the mesodermal layer as cardiomyocytes. Although we did not show cardiomyocyte differentiation from the same patient, due to the ethical issue, we showed that hGFCs could be differentiated into cardiomyocytes in vitro.

Patients with CAD can consequently suffer from ischemic cardiomyopathy and congestion heart failure. The progression of the disease will cause the failure of medical treatment and thus require a more advanced treatment. Moreover, the occurrence of upper GI bleeding is high due to antiplatelet treatment [28,29]. GI endoscopy examination is also necessary for the diagnosis and treatment of gastric ulcer bleeding. If an ulcer is present, a biopsy will also be performed simultaneously [30]. iPSCs have provided great potential for this issue. Matsumoto et al. used human iPSC-derived cardiomyocytes and endothelial and smooth muscle cells to construct cell sheets in vitro. Then, the cell sheets were transplanted into a rat myocardial infarction model. They showed that the heart had functional and electrical recovery 2 weeks after transplantation [31]. Recently, researchers in Japan have performed a clinical trial in which they used iPSC-generated cardiomyocytes to construct cell sheets and transplanted them onto a damaged myocardium [32]. We observed that the hGFCs obtained from GI endoscopy biopsy can be generated fruitfully and provide an accessible source to generate iPSCs. hGFC-derived iPSCs are morphologically similar to ESCs and express specific surface markers of stem cells. Teratoma formation was shown in the in vivo tissues containing all three germ layers, including ectoderm, mesoderm, and endoderm (Figure 4). Further, these hGFC-derived iPSCs differentiated into functional cardiomyocyte-like cells that expressed cardiomyocyte markers (Figure 5). This study may provide a novel cell source and strategy for iPSC-personalized therapies for human disease in the future.

## 5. Conclusions

To our knowledge, the present study is the first report of IPSCs derived from hGFCs that were obtained from PES biopsy. In summary, we established a protocol to obtain hGFCs and generate iPSCs via tiny gastric tissue collected from PES with biopsy. These hGFCs can be differentiated into cardiomyocytes. Using the proposed method, we may create a hGFC-derived iPSC bank for patients who have cardiovascular disease coinciding with gastric disease. This study also provides future perspectives for hGFC-derived iPSCs and the development of personalized regenerative therapy.

## Figures and Tables

**Figure 1 biomedicines-09-01565-f001:**
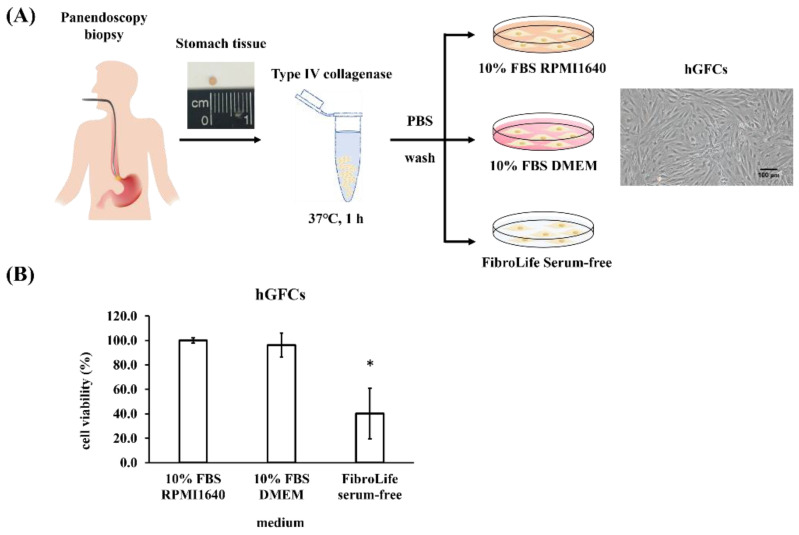
Isolation and characterization of hGFCs. (**A**) Schematic of the isolation protocol used. The stomach tissue was digested with collagenase IV for 1 h. The cells were cultured with 10% FBS RPMI1640, 10% FBS DMEM, or FibroLife serum-free medium, and images of cell morphology were obtained by a digital microscope. Scale bar 100 μm (100× magnification). (**B**) hGFCs were seeded and cultured with 10% FBS RPMI1640, 10% FBS DMEM, or FibroLife serum-free medium for 72 h. The WST-1 assay was performed to measure the cell viability. This experiment was performed in triplicate, and data are presented as mean ± SD. * *p* < 0.05 compared with 10% FBS RPMI1640.

**Figure 2 biomedicines-09-01565-f002:**
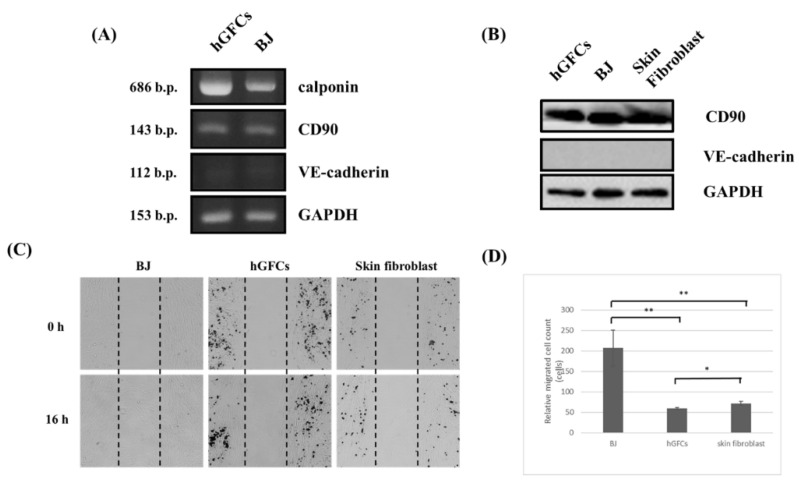
Analysis of the character of fibroblasts. The fibroblasts from stomach tissue (hGFCs) and BJ cells were collected, and the expressions of fibroblast markers calponin and CD90 and epithelium marker VE-cadherin were analyzed by (**A**) RT-PCR. In addition, the protein expressions of hGFCs, BJ cells, and skin fibroblasts were analyzed by Western blot. (**B**) The expression of fibroblast marker CD90 is presented, and none of epithelium marker VE-cadherin. (**C**) The wound healing assay was performed to investigate the migration of fibroblasts. (**D**) The data revealed that cell migration was slow in hGFCs compared with BJ cells and skin fibroblasts. Images were acquired at 40X. The hGFCs reveal less migration activity than skin fibroblasts and BJ cells. The experiment was performed in triplicate, and data are presented as mean ± SD. * *p* < 0.05, ** *p* < 0.01.

**Figure 3 biomedicines-09-01565-f003:**
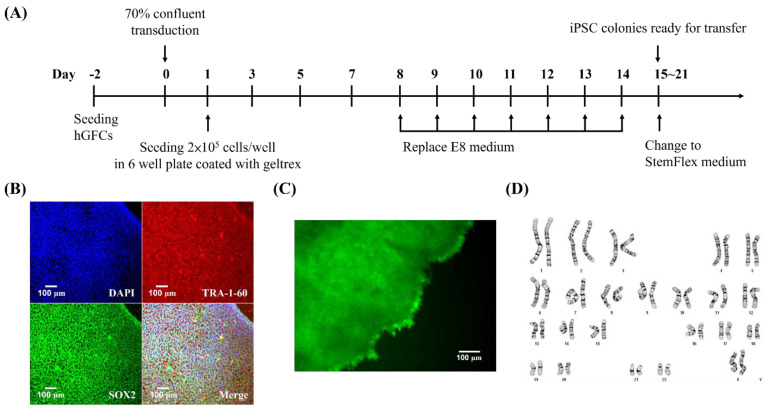
Generation of iPSCs from hGFCs. (**A**) Schematic of the protocol of generating iPSCs from hGFCs. Primary hGFCs were reprogrammed with transcription factors, including OCT3/4, SOX2, KLF4, and c-Myc. (**B**) The reprogrammed iPSCs expressed the specific pluripotency markers, namely, TRA-1-60 (red) and SOX2 (green). Scale bar 100 μm (100× magnification). (**C**) The features of hGFC-derived iPSCs were analyzed by ALP staining. The images of ALP expression (green) were captured by a digital microscope. Scale bar 100 μm (100× magnification). (**D**). Karyotyping and G-band analysis revealed a normal karyotype (46,XX).

**Figure 4 biomedicines-09-01565-f004:**
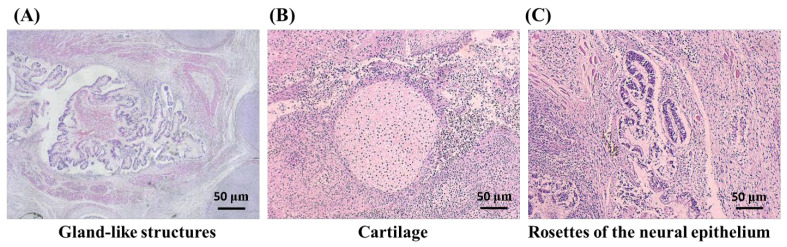
Histological examination of teratomas from hGFC-derived iPSCs. hGFC-derived iPSCs were injected into mice subcutaneously. After the tumors formed, histological examination was performed by H&E staining. The images of teratomas composed of three germ-layer tissues presented (**A**) gland-like structures, (**B**) cartilage, and (**C**) rosettes of the neural epithelium. Scale bars 50 µm (40× magnification).

**Figure 5 biomedicines-09-01565-f005:**
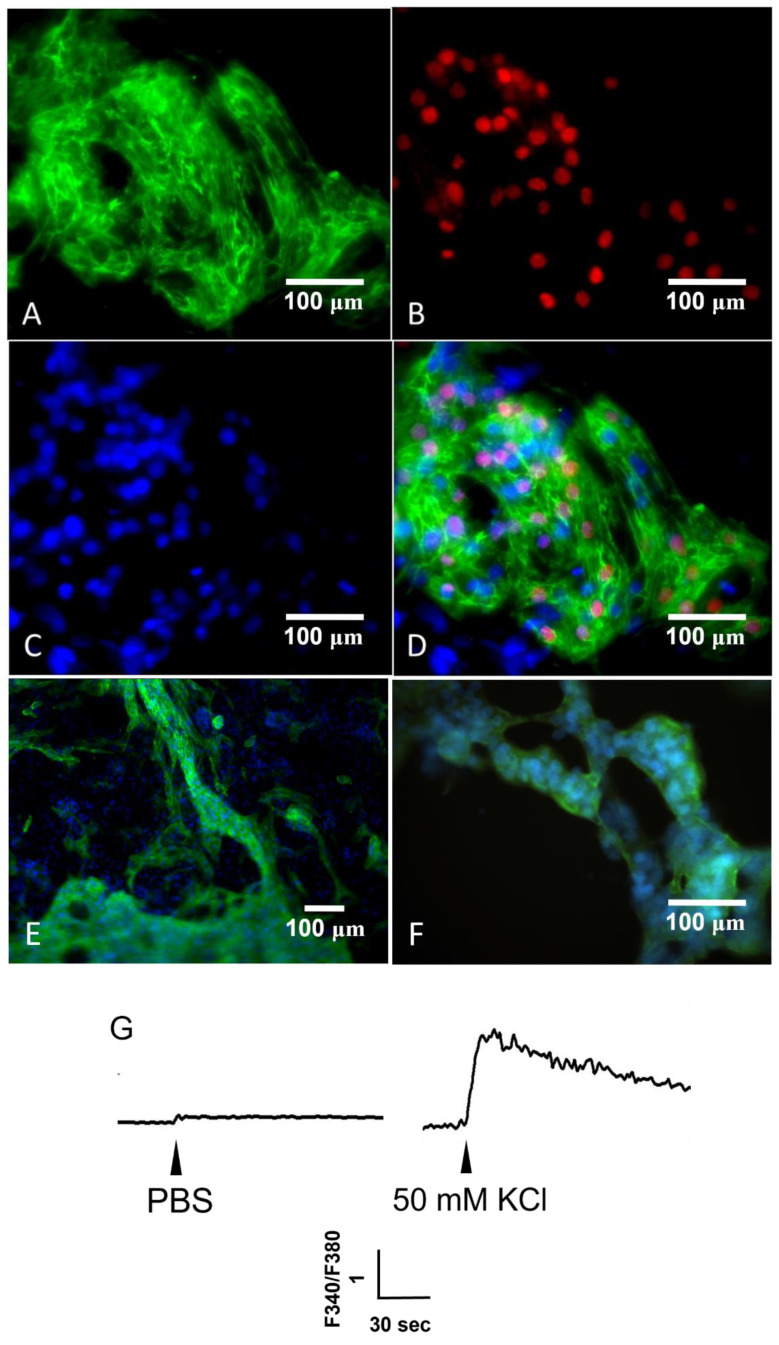
Characteristics of iPSC-differentiated cardiomyocytes. The iPSCs were cultured with the cardiomyocyte differentiation medium followed by the maintenance medium. After 14 days, the cardiac markers of cells, namely, (**A**) TNNT2/cTNT (green) and (**B**) NKX2.5 (red), were analyzed by immunofluorescence staining. (**C**) DAPI was used as the nuclear DNA stain (blue). (**D**) The merged image is presented. Scale bar 100 μm, (200× magnification). In addition, another two cardiomyocyte-specific markers, (**E**) alpha-actinin (Scale bar 100 μm, 100× magnification) and (**F**) MYL2 (Scale bar 100 μm, 200× magnification) were also analyzed. (**G**) Fura-2 loaded cells grown on coverslips were bathed in Locke’s buffer. The fluorescence intensities of single cells were measured as an index of the change in cytosolic Ca^2+^ levels (F340/F380) in response to the addition of phosphate-buffered saline (PBS) (left trace) or 50 mM KCl (right trace), as indicated by arrowheads.

## Data Availability

All data are available within the manuscript and upon request to corresponding author.

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
