# Peer review of "Generation of Functional Cardiomyocytes from Human Gastric Fibroblast-Derived Induced Pluripotent Stem Cells"

_biomedicines, 2021, doi:10.3390/biomedicines9111565_

Round 1

Reviewer 1 Report

This paper describes the generation of cardiomyocytes from human iPSCs. I found more details of characterization of differentiated cardiomyocytes could be really helpful to assess whether the iPSC-derived cardiomyocytes actually can mimic those cells in vivo. Only two markers (NKX2.5 and TNNT2/cTNT) are not enough. 

Author Response

Dear Professor and Ms. Liao,

Thank you for giving me the opportunity to submit a revised draft of my manuscript titled: “Cilostazole induces eNOS and TM expression via activation with Sirtuin 1/ Krüppel-like factor 2 pathway in endothelial cell” to International Journal of Molecular Sciences. I appreciate the time and effort that you and the reviewers have dedicated to providing your valuable feedback on my manuscript. I am grateful to the reviewers for their insightful comments on my paper. I have been able to incorporate changes to reflect most of the suggestions provided by the reviewers. I have highlighted the changes within the manuscript.

Here is a point-by-point response to the reviewers’ comments and concerns.

Reviewer 2 Report

In this paper, we established a protocol for human gastric fibroblast isolation using collagenases and inducible pluripotent stem cell (iPSC) generation from gastric fibroblast via PES with biopsy. Finally, the authors showed that these iPSCs can be differentiated into beating cardiomyocytes in vitro. Although their experiment was well performed, this paper lacks for the conceptual progress in the field. Currently, iPS technology is widely used in different field of research and it is not novel with the fact that they created functional cardio myocyte from fibroblast. Therefore, I judge it is not recommendable for the publication in Biomedicine unless they have new findings with their established cells (applications and advantages compare to normal fibroblast cells).

Author Response

Dear Professor and Ms. Li,

Thank you for giving me the opportunity to submit a revised draft of my manuscript titled: “Generation of beating cardiomyocytes from human gastric fibroblast derived induced pluripotent stem cells” to Biomedicines. I appreciate the time and effort that you and the reviewers have dedicated to providing your valuable feedback on my manuscript. I am grateful to the reviewers for their insightful comments on my paper. I have been able to incorporate changes to reflect most of the suggestions provided by the reviewers. I have highlighted the changes within the manuscript.

Here is a point-by-point response to the reviewers’ comments and concerns.

Reviewer 3 Report

Chih-Hsien Wu and colleagues in their studies adopted a method to generate human induced pluripotent stem cells (hiPSCs) from patient gastric fibroblasts cells (hGFCs) in standard feeder-free conditions and with the use of a commercially available reprogramming kit.  The authors showed that the developed hiPSCs are pluripotent and can be successfully differentiated to cardiomyocytes.  The manuscript is a short report describing an availability of stomach biopsy samples as source of gastric fibroblasts cells which potentially can replace human dermal fibroblasts in iPSC-generation.  The method of iPSC generation from human dermal fibroblast is well established and extensively used in current research and clinical practices. Potentially hGFCs-derived iPSC can be used to get differentiated cells of interest for purposes of regenerative medicine, diseases modeling and drug development. Due to small volume of research data, this paper can be potentially accepted for publication in Biomedicines as a short method report.

The title should be modified – remove “beating”, or replace it to “functional”. However, there are not enough data in the paper supporting that the generated fibroblasts are functional. There are only two cardiomyocyte markers used and a supplementary movie file showing some pulsation of the cell culture. It is better to perform more rigorous investigation of functioning the derived cardiomyocytes.

The abstract is pretty short, that reflects the small volume of the paper. First sentence of the abstract is confusing why these two diseases are together and how their etiology is linked. These are different diseases and a main focus of this study is directed to development of potential cardio pathology treatment, to this regard a coronary artery disease may be mentioned but peptic ulcer should be removed from abstract and in some part of the introduction. Replace “inducible” to “induced” in the title and in the text. In introduction and discussion sections, the statements - that trypsinization is damaging for fibroblast is overstated and is not relevant to this study. This should be explained and modified accordingly.

In the introduction and in the discussion, sections should be clearly explained what are advantages to use hGFCs over the use of human dermal fibroblasts in iPSCs generation.

Should be explanations for BJ cells, no word in the paper about what kind of these cells.

It is important to compare primary hGFCs with human primary dermal fibroblasts not with BJ cells.

WST-1, TNNT2 abbreviations should be explained.  

English should be improved.

Author Response

(The authors gave the same response as above.)

Round 2

Reviewer 2 Report

To whom it may concern

The authors are not providing the novelty of their study but just describe about the new result. They should impact on the novelty of their study. Based on what they described in their response, I cannot see it. 

Author Response

Dear Professor and Ms. Li,

Thank you for giving me the opportunity to submit a revised draft of my manuscript titled: “Generation of beating cardiomyocytes from human gastric fibroblast derived induced pluripotent stem cells” to Biomedicines. I appreciate the time and effort that you and the reviewers have dedicated to providing your valuable feedback on my manuscript. I am grateful to the reviewers for their insightful comments on my paper. I have been able to incorporate changes to reflect most of the suggestions provided by the reviewers. I have highlighted the changes within the manuscript.

Here is a point-by-point response to the reviewers’ comments and concerns.

Comments from Reviewer 2

Comment 1: The authors are not providing the novelty of their study but just describe about the new result. They should impact on the novelty of their study. Based on what they described in their response, I cannot see it.

Response: Thank you for pointing this out.

To impact on the novelty of our study, we had emphasized the novelty in this revised drafts.

  1. Established a new hGFCs isolation protocol from the tiny residual gastric biopsy samples:

   The panendoscopy is always performed to detect the peptic ulcer and treat ulcer bleeding. Then the gastric biopsy will be done to exclude the possible gastric cancer and Helicobacter pylori infection. Some gastric samples would be wasted after the assay was completed. This inspired us the ideal to using to isolate and culture gastric fibroblasts from these residual samples. At present, most of gastric cell isolations were collected from gastric cancer patients who underwent gastrectomy. The cells may contain characteristics of gastric cancer, which limit its use in clinical settings for future iPSC therapy and personalize cell bank set up.

To correct vague discerptions, we had made a change in the introduction section:

In introduction section: Page 2, line 18-19 of: the sentence “Collagenases are applied as suitable enzymes for the separation of tissue cells in medical investigation, especial for tiny-tissue cell collection. In the present study, we established a protocol for isolated hGFCs and their iPSC generation. “ had been modified to “Collagenases are applied as suitable enzymes for the separation of tissue cells in medical investigation, especial for tiny-tissue cell collection. In the present study, we established a new protocol for isolated hGFCs from tiny gastric tissue and their iPSC generation.”.

  1. Established a novel reprogram protocol for hGFCs derived IPSCs generation.

   The fibroblasts retain the de-differentiation potential and pluripotent properties which could enhance reprogramming capacity (2). It suggests the fibroblasts remain prime candidates to establishing induced pluripotent stem cells (iPSCs). However, the tissue residual fibroblast originated from different gremial layers could have different plasticity (3). In addition, Sanchez-Freire et al had reported the different somatic cells could lead to different capacity to differentiate to specific cells. The study suggested that while epigenetic memory will impact the differentiation efficiency of somatic cell source. The skin is derived from ectodermal layer. However, the gastric connective tissue originated from mesodermal layer which as cardiomyocytes. In addition, the different somatic cell might need different protocol to generate IPSCs (4,5,6). In our best knowledge, this is the first case of primary cultured hGFCs could be generated into IPSCs with novel reprogram protocol.

To correct vague discerptions, we had made a change in the abstract section:

In abstract section: Page 1, line 7-8 of abstract section: Added the sentence “In our knowledge, this is a first case to generate iPSCs from gastric fibroblast in vitro. “.

Comment: English should be improved.

Response: Thank you for pointing this out. We had re-edited the English by Dr Chih-Hsien Wu.

Reference:

  1. Ma, Y., Zhu, J., Chen, S., Li, T., Ma, J., Guo, S., Hu, J., Yue, T., Zhang, J., Wang, P., Wang, X., Chen, G., and Liu, Y. (2018) Activated gastric cancer-associated fibroblasts contribute to the malignant phenotype and 5-FU resistance via paracrine action in gastric cancer. Cancer Cell Int 18, 104
  2. Takahashi, K., Tanabe, K., Ohnuki, M., Narita, M., Ichisaka, T., Tomoda, K., and Yamanaka, S. (2007) Induction of pluripotent stem cells from adult human fibroblasts by defined factors. Cell 131, 861-872
  3. LeBleu, V. S., and Neilson, E. G. (2020) Origin and functional heterogeneity of fibroblasts. FASEB J 34, 3519-3536
  4. Takahashi, K., Tanabe, K., Ohnuki, M., Narita, M., Ichisaka, T., Tomoda, K., and Yamanaka, S. (2007) Induction of pluripotent stem cells from adult human fibroblasts by defined factors. Cell 131, 861-872
  5. Takahashi, K., and Yamanaka, S. (2006) Induction of pluripotent stem cells from mouse embryonic and adult fibroblast cultures by defined factors. Cell 126, 663-676.
  6. Zhou, T., Benda, C., Duzinger, S., Huang, Y., Li, X., Li, Y., Guo, X., Cao, G., Chen, S., Hao, L., Chan, Y. C., Ng, K. M., Ho, J. C., Wieser, M., Wu, J., Redl, H., Tse, H. F., Grillari, J., Grillari-Voglauer, R., Pei, D., and Esteban, M. A. (2011) Generation of induced pluripotent stem cells from urine. J Am Soc Nephrol 22, 1221-1228

Reviewer 3 Report

Important to go through the text of the article and thoroughly make correction check.

AHA and ESC abbreviations require explanation.

Alpha-actin

Please remove all old images on the modified version of the manuscript.

Figure 2(C) – the experiment is not informative or performed incorrectly, α-SMA is absent in all samples, however it presents in base on immunocytochemistry Figure 2(B). CD90 is absent in absent in skin fibroblasts, why? The panel B should be removed if no explanation or this experiment should be redone with other antibodies etc.

“The cells from stomach tissue”, should be - The fibroblasts from stomach tissue (abbreviation), explain in the text that at the used culture condition the only fibroblasts can survive.

Please correct some parts of the figure 2 legend.

The complete evidence for functionality of iPSC-derived cardiomyocytes is not presented in the paper.

Author Response

Dear Professor and Ms. Li,

Thank you for giving me the opportunity to submit a revised draft of my manuscript titled: “Generation of beating cardiomyocytes from human gastric fibroblast derived induced pluripotent stem cells” to Biomedicines. I appreciate the time and effort that you and the reviewers have dedicated to providing your valuable feedback on my manuscript. I am grateful to the reviewers for their insightful comments on my paper. I have been able to incorporate changes to reflect most of the suggestions provided by the reviewers. I have highlighted the changes within the manuscript.

Here is a point-by-point response to the reviewers’ comments and concerns.

Comments from Reviewer 3

Comment: AHA and ESC abbreviations require explanation. Alpha-actin

Response: Thank you for pointing this out. The abbreviations AHA, ESC and Alpha-actin had been explained in the introduction section. The typo error of alpha-actin had been corrected in the material method, result and figure 5 legend section.

Comment: Please remove all old images on the modified version of the manuscript.

Response: Thank you for pointing this out. The all old images had been removed on the modified version.

Comment: Figure 2(C) – the experiment is not informative or performed incorrectly, α-SMA is absent in all samples, however it presents in base on immunocytochemistry Figure 2(B). CD90 is absent in absent in skin fibroblasts, why? The panel B should be removed if no explanation or this experiment should be redone with other antibodies etc.

Response: Thank you for pointing this out and suggestion. Indeed, our data reveals very week expression of α-SMA in western blot. It will make confuse of the result. The panel B had been removed. In addition, we had performed the western blot for CD90 detection again to verify the expression of CD 90 in three kinds of fibroblast cells. We had adjusted the western blot expression in Figure 2 (C). 

Comment: The cells from stomach tissue”, should be - The fibroblasts from stomach tissue (abbreviation), explain in the text that at the used culture condition the only fibroblasts can survive.

Response: Thank you for pointing this out. The cells from stomach tissue” had been corrected to “The fibroblasts from stomach (hGFCs)” in figure 2 legend. To explain the culture condition the only fibroblast can survive. We add the sentence in result section: Page 5, line 6-7: modified the previous sentence “The cells presented a typical spindle shape, and cells from passages 6 to 12 were used for subsequent studies”  to  “The cells presented a typical spindle shape, and cells from passages 6 to 12 were used for subsequent studies, due to the others kinds cells would not survive in this culture condition and time point (8)” .

Comment: Please correct some parts of the figure 2 legend.

Response: Thank you for pointing this out. The figure 2 legend had been rewritten.

Comment: The complete evidence for functionality of iPSC-derived cardiomyocytes is not presented in the paper.

Response: We appreciate your comment and sorry for the lack of the functional test of iPSC-derived cardiomyocytes in the previous revised draft. To correct this and investigate the functionality of iPSC-derived cardiomyocytes, we measured the intracellular Ca2+ change in response to 50 mM KCl treatment which activates L type Ca2+ channels. Marked Ca2+ increased was induced by high KCl, while it remained unaltered by phosphate buffered saline which as normal cardiomyocyte present (Figure 5G).

Figure 5 (G) legend:

Fura-2 loaded cells grown on coverslips were bathed in Locke's buffer. The fluorescence intensities of single cells were measured as an index of the change in cytosolic Ca2+ levels (F340/F380) in response to the addition of phosphate buffered saline (PBS) (left trace) or 50 mM KCl (right trace), as indicated by arrowheads.

Comment: English language and style are fine/minor spell check required.

Response: Thank you for pointing this out. We had rechecked the English spell by Hsuan-Hwai Lin.
